# Finite Element Analysis of Fixed Orthodontic Retainers

**DOI:** 10.3390/bioengineering11040394

**Published:** 2024-04-18

**Authors:** Sebastian Hetzler, Stefan Rues, Andreas Zenthöfer, Peter Rammelsberg, Christopher J. Lux, Christoph J. Roser

**Affiliations:** 1Department of Prosthodontics, University of Heidelberg, Im Neuenheimer Feld 400, 69120 Heidelberg, Germany; 2Department of Orthodontics and Dentofacial Orthopedics, University of Heidelberg, Im Neuenheimer Feld 400, 69120 Heidelberg, Germanychristoph.roser@med.uni-heidelberg.de (C.J.R.)

**Keywords:** retainer stiffness, tooth resilience, bite force distribution, adhesive failure

## Abstract

The efficacy of retainers is a pivotal concern in orthodontic care. This study examined the biomechanical behaviour of retainers, particularly the influence of retainer stiffness and tooth resilience on force transmission and stress distribution. To do this, a finite element model was created of the lower jaw from the left to the right canine with a retainer attached on the oral side. Three levels of tooth resilience and variable retainer bending stiffness (influenced by retainer type, retainer diameter, and retainer material) were simulated. Applying axial or oblique (45° tilt) loads on a central incisor, the force transmission increased from 2% to 65% with increasing tooth resilience and retainer stiffness. Additionally, a smaller retainer diameter reduced the uniformity of the stress distribution in the bonding interfaces, causing concentrated stress peaks within a small field of the bonding area. An increase in retainer stiffness and in tooth resilience as well as a more oblique load direction all lead to higher overall stress in the adhesive bonding area associated with a higher risk of retainer bonding failure. Therefore, it might be recommended to avoid the use of retainers that are excessively stiff, especially in cases with high tooth resilience.

## 1. Introduction

Fixed orthodontic retainers are needed to stabilise the alignment but are only sufficient if they are successful in the long term. Non-sufficient fixed retainers can cause orthodontic relapse [1] and removing a failed retainer can damage tooth enamel [2]. Complications with fixed retainers are particularly troublesome for the orthodontist, as active treatment is usually finished by the time the retainer fails. Retainer failure might therefore stay undetected until the teeth begin to move, resulting in an orthodontic relapse requiring new orthodontic treatment.

Retainer failure rates are relatively high, with 35.2% of conventional (bendable) fixed retainers failing [3], and these rates might even be higher in fixed retainers fabricated by computer-aided design/manufacturing (CAD/CAM) [4]. How differences in biomechanical behaviour affect the sufficiency of fixed retainers remains relatively unknown. Young’s modulus (E), retainer diameter (RD), and retainer configuration (multistranded, braided, flat, etc.) might be the most important factors because they influence retainer (bending) stiffness (RS) and thus determine the bite force that is transmitted from a loaded tooth to the neighbouring teeth. Consequently, stresses occurring along the adhesive bonding interfaces might therefore be affected by the material and geometry of the retainer.

There are considerable differences between fixed retainers, and these differences affect their biomechanical behaviour. For example, fixed retainers have different designs, materials, and fabrication techniques. In the case of commercially available conventional fixed retainers, the outer diameter can range from 0.38 mm (0.015 inches) to 0.81 mm (0.032 inches), and new fixed CAD/CAM retainers can reach 3.5 mm (0.138 inches) in height. Retainer geometry can also differ; conventional fixed retainers can be flat, multistranded, or braided, whereas CAD/CAM retainers have a highly variable geometry. Regarding the retainer material, conventional fixed retainers are made from stainless steel, gold, grade 1 titanium, grade 5 titanium, or titanium-molybdenum, whereas fixed CAD/CAM retainers can also be made from polyetheretherketone, zirconia (ZrO_2_), nickel titanium, or cobalt-chromium. The material selection directly influences the RS due to the resulting differences in Young’s modulus and might therefore affect the transmission of force from a loaded tooth to the neighbouring teeth.

Finite element (FE) analysis is an important part of research in many branches of dentistry. Examples include the field of implantology [5], craniomaxillofacial surgery [6], endodontics [7], prosthetics [8], material research [9], and also of orthodontics [10]. Several studies have investigated retainer failure using finite element (FE) analysis [11,12,13] and have analysed their bonding behaviour [14,15]. However, little is known about how RS affects the biomechanical behaviour of teeth bonded by a fixed retainer. These effects might be important, particularly for newer CAD/CAM retainers. Previous in vitro investigations have shown that an increasing RS significantly decreases tooth mobility [16] and significantly increases failure rates [4].

Herein, we used an FE model of a lower jaw anterior segment to investigate the effects of RS on the transmission of the force and distribution of stress along adhesive interfaces. We concentrated on the enamel–adhesive interface, because detachment in this interface is the most common failure [3]. The FE model was developed to understand why different retainers have different failure rates. We also investigated the influence of tooth resilience (TR) by simulating different patient situations.

## 2. Materials and Methods

Before constructing the FE model, we determined the range of RS of different commercially available multistranded fixed retainers using standardised experimental testing. Due to their complex geometry, such testing was not feasible for fixed CAD/CAM retainers. Therefore, an approximation of the RS based on its dimensions (1.5 × 3.5 mm) was made for a fixed CAD/CAM retainer made from ZrO_2_ (Zahnwerkstatt Wernigerode, Germany).

### 2.1. Bending Stiffness of Multistranded Retainers

Exemplary conventional hand-bent retainers were tested in a three-point bending test (according to DIN EN ISO 15841 [17]) to put the results gained from the FE simulation into a clinical context. The test setup is displayed in Figure 1. All the retainers are listed in Table 1.

In the following FE computations, multistranded retainers were simplified as cylindrical wires. The equivalent Young’s modulus (*E**) of such a cylindrical wire with the same diameter (*RD*) as the multistranded retainer and showing the same RS (*k*) with respect to bending can be calculated as follows:(1)$$E^{*}=\frac{k\times L^{3}}{48\times I}$$
(2)$$I=\frac{\pi \times {RD}^{4}}{64}$$
with *I* being the geometrical moment of inertia for a circular cross section. The bending stiffness (*k*) was calculated based on in vitro tests (*n* = 5) as described above with *L* = 10.15 mm as a regression line between 0.1 mm and 0.5 mm displacement.

### 2.2. Finite Element Model

Based on the anterior segment of a typodont lower jaw model (ANA-4, Frasaco, Tettnang, Germany), an FE model was created from the left canine to the right canine (tooth 33 to tooth 43, FDI scheme) using hexahedral elements with quadratic shape functions (ANSYS R22, CADFEM, Canonsburg, PA, USA). Approximal tooth–tooth contacts were simulated with area-to-area contact elements using the penalty method (Figure 2).

The same typodont model geometry has been transferred to 3D-printed in vitro models for retainer testing in previous studies [4,16]. Analyses based on this FE model were intended to generate further insight into the outcome of these investigations.

#### 2.2.1. Tooth Resilience

The FE investigation concentrated on the resulting TR in the vertical and horizontal direction; therefore, uniform and simplified cylindrical geometries (Figure 2a) were used for the dentine root and periodontal ligament. By varying the material properties of the periodontal ligament, it was possible to achieve low (1 µm/N), medium (2 µm/N), or high (3 µm/N) TR values in the vertical direction. Previous studies [18,19] have indicated that the ratio between the horizontal and vertical TR should lie between three and four. Detailed information about the simulated TR is presented in Table 2.

A consistent TR was selected for each calculation across all teeth. The nodes of each periodontal ligament cylinder were completely restricted at the circumferential outer surface, so no bone structures had to be implemented in the FE model.

#### 2.2.2. Retainer Design, Attachment, and Loading Conditions

A retainer with a circular cross section was added on the oral side as specified by an orthodontist and connected to each tooth with a bonding gap of 0.1 mm between the tooth surface and wire. The bond was realised by an adhesive material implemented via cylindrical bonding spots with a diameter of 2.5 mm. The height of the cylinders was chosen such that the minimum oral coverage of the wire with adhesive was 0.4 mm for the canine teeth and 0.2 mm for all the other teeth. The material parameters of the model are given in Table 3.

To easily compare the results gained from the different combinations of *E* and *RD*, all the retainer bending stiffnesses were normalised by the least stiff retainer (*k*_0_) to give the normalised bending stiffness (*k*/*k*_0_). The list of the simulated retainers, ordered by increasing bending stiffness, is given in Table 4.

To ensure easy readability, the relative bending stiffnesses (*k*/*k*_0_) will be used to refer to the individual retainers throughout this investigation. Additional FE computations were carried out with the largest *RD* and an extremely high Young’s modulus (enlarged by 10^6^) to simulate the effect of a rigid retainer.

A cross section of the FE model with the teeth, adhesive (composite resin cylinders), retainer, and periodontal ligament given for a single tooth, with the two different load cases, is displayed in Figure 2a. The anterior segment of the lower jaw with the least and most stiff retainer is illustrated in Figure 2b.

A bite force of 100 N was evenly distributed to the edge of the left central incisor (tooth 31) with two different load cases (LC): axial (LC1) or oblique (LC2, 45° tilt to the labial side with a lever arm of approximately 12 mm in length).

### 2.3. Analysed Data

The amount of bite force transmitted to the teeth next to the loaded tooth via its adhesive bonding area and the stress distribution at the bonding interfaces were examined. To get a clear display of the effects of the RS, TR, and LC, the relative force transmission was plotted over the relative retainer bending stiffness. An exponential function of the type presented by Formula (3) correlating the relative transmitted force (ratio of the transmitted force and the applied resultant force, *F_trans_*/*F_res_*) and RS (*k*) using an arbitrary reference stiffness (*k*_0_) (least stiff retainer) as well as parameters a and b was suited to fit the *FE* computation data as it shows asymptotic behaviour towards an upper (*F_rel,rigid_*, relative force transmission for a rigid retainer) and lower threshold (no force transmission if the retainer has no stiffness).
(3)$$\frac{F_{trans}}{F_{res}}=F_{rel,rigid}\times e^{-\frac{a}{{\left(\frac{k}{k_{0}}\right)}^{b}}}$$
$$\frac{F_{trans}}{F_{res}}\;\to 0\;for\;k\to 0$$
$$\frac{F_{trans}}{F_{res}}\to F_{rel,rigid}\;for\;k\to \infty$$

To calculate whether the adhesive bond is susceptible to debonding and to illustrate the stress distribution at the bonding interfaces, the normal (*σ*) and resulting shear stresses (*τ*) in the bonding interfaces corresponding to a bite force of 10 N were calculated and implemented using a fracture hypothesis (Formula (4)) and typical values for the tensile bond strength ($\sigma_{u}$ = 20 MPa) and shear bond strength (*τ_u_* = 20 MPa) for the interface between the teeth and the adhesive [20,21,22,23]. The resulting value represents the utilised capacity of the adhesive bond for the corresponding combination of the RS, TR, and load case. A high degree of utilised capacity in a certain area therefore corresponds to high normal and resulting shear stresses in that area.
(4)$$\mathrm{No}\;\mathrm{debonding}\;\mathrm{for}\;\frac{\sigma }{\sigma_{u}}+\frac{\tau }{\tau_{u}}<1$$
$$\mathrm{Debonding}\;\mathrm{for}\;\frac{\sigma }{\sigma_{u}}+\frac{\tau }{\tau_{u}}=1$$

## 3. Results

### 3.1. Experimental Testing

The results of the three-point bending tests are listed in Table 5.

Multistranded retainers had a 10–15 times lower RS than solid wires with the same diameter. This can be seen from the adapted Young’s moduli presented in Table 4 for multistranded retainers. Because the RS correlates to the *RD* with the power of 4 (Formulas (1) and (2)), a rapid increase in stiffness can be seen with an increasing *RD*.

### 3.2. Force Transmission to the Bonding Area

Force transmission to the retainer via the bonding area increased with increasing TR and RS for both load cases (see Figure 3). With the highest TR and RS, about two-thirds (65.1% for LC1 and 69% for LC2) of the bite force was transmitted to the bonding area of the loaded tooth, making it the most susceptible for debonding. In contrast, with the lowest TR and RS, only about 2% for LC1 and 7% for LC2 of the bite force were transmitted.

Within commercially available conventional hand-bent retainers (green rectangle, displayed with yellow lines in Figure 3), the differences in the relative force transmission were up to 25%. The force transmission to the bonding area was lowest for the 0.038 mm, six-strand twisted stainless steel retainer (no. 4), which corresponds to the retainer with the overall lowest RS (*k*/*k*_0_ = 1). The force transmission values within this group were the highest for the 0.081 mm, six-strand twisted stainless steel retainer (no. 2; about 27% and 36% for LC1 and LC2, respectively). The force transmission values for all the other conventional retainers were in between. The ZrO_2_ CAD/CAM retainers (no. 6) were included as an extreme example of retainers with a high RS and showed force transmission values up to 4.8 times higher than those for the stiffest conventional retainer (at low TR). Applying an oblique bite force increased the force transmission, making this load case more critical for debonding.

Increasing the RS by a factor of 10^6^ increased the relative force transmission to the upper threshold, corresponding to a complete rigid retainer, which amounted to *F_rel,rigid_* = 79.4% for LC1 and *F_rel,rigid_* = 90.1% for LC2 (see Formula (3)). The fitted parameters a and b for the exponential function given in Formula (3) are listed in Table 6.

### 3.3. Effect of Retainer Diameter on Stress Distribution

While the overall amount of force transmitted to the bonding area of the loaded tooth increased with increasing RS, the utilised capacity calculated with Formula (4) did not necessarily increase as well because of the non-uniformity of the stress distribution (Figure 4 and Figure 5). Figure 4 presents the maximum degree of utilised capacity for selected retainers that illustrate the effects of an increasing RS due to the increase in *RD* and/or the use of a solid cross section instead of a multistranded configuration. The resulting value corresponds to the highest amount of utilised capacity that occurs at each tooth, respectively. Within the region of conventional hand-bent retainers (comparing *k*/*k*_0_ = 1 and *k*/*k*_0_ = 12.2), the increase in RS (here by a factor of 12.2) increased the utilised capacity. Although this was the case for both the increase in RD (dashed blue curve in Figure 4) and a switch to the solid cross section (solid blue curve), the maximum degree of utilised capacity increased less with the increase in *RD*. This effect of a more uniform stress distribution and therefore the distribution of the utilised capacity with a thicker retainer was more prominent with a higher RS. Comparing the dashed blue line with the dashed red line, the RS and accordingly the force transmission increased only by a rising *RD*. However, the maximum utilised capacity at tooth 31 decreased because the stress distribution was more uniform. This effect can be seen in Figure 5c. In contrast, by switching to a solid cross section (solid red curve) instead of changing the *RD*, the maximum utilised capacity increased slightly. Because the force transmission via the adhesive bond was nearly identical for both retainers, it was clear that the less-uniform stress distribution for the thinner retainer caused this effect. A good example of such a non-uniform stress distribution can be found in the solid blue curve in Figure 4 and in Figure 5b.

Figure 6 shows that switching from LC1 to LC2 clearly shifted the maximum degree of the utilised capacity to the neighbour teeth of the loaded tooth, especially to tooth 32. The increase in TR slightly increased the maximum degree of the utilised capacity while its distribution pattern stayed the same.

## 4. Discussion

The present study demonstrates that the main parameters affecting force transmission and stress distribution in the retainer-adhesive system are the RS and TR. With regard to the RS, both the RD and its configuration significantly influenced the stress distribution. Decreasing the RD contributed to a less-uniform stress distribution, which in turn contributed to high local stress peaks. However, these were limited to a very small area. Vice versa, increasing the RS (especially by increasing the diameter) led to a more uniform stress distribution but increased the overall stress on the bonding system, making fixed retainers with higher stiffness more susceptible to bonding failure. Regarding the load direction, oblique bite forces increased the force transmission to neighbouring teeth, making this load case more critical for possible debonding. Moreover, the chosen exponential function (Formula (3)) gave a good fit of the results. These results are important because they visually indicate the different force transmission and stress distribution patterns and support the clinician when choosing the respective retainer for the individual patient in order to guarantee low complication rates.

The force transmission and stress distribution could only be visualised by the FE model presented here, which is a strength of the study. The model adequately simulated how the RS affects force transmission, improving our understanding of retainer failure. The model also simulated various bite situations, with two representatives (LC1 and LC2). Through the variable adjustment of the TR, different patient situations and the conditions of thematically related in-vitro studies [24] could be simulated as well. Thus, it was possible to simulate a particularly critical clinical case of unrestricted tooth mobility for the loaded tooth while keeping the neighbour teeth completely rigid. When comparing the results for the different TR adjustments, differences in the stiffness between the lowest and highest TR of up to a factor of 3.6 were observed. This clearly shows that implementing TR is necessary to accurately simulate the clinical situation. The FE analysis allowed a wide range of RS values to be included instead of limiting the investigation to a few commercially available fixed retainers. This might be particularly relevant for developing new fixed retainers, including newly upcoming fixed CAD/CAM retainers, which are made from various materials like polyetheretherketone, cobalt-chromium, or ZrO_2_ and have various designs that have never been used to produce hand-bent fixed retainers before. The ZrO_2_ retainer was included as an example of a fixed retainer with a particularly high RS, and the relative force transmission of this retainer was up to 4.8 times higher than that of the stiffest conventional fixed retainer. From a clinical point of view, such higher stiffness values, which correspond to higher force transmission and higher overall stress values (as long as the adhesive bonding area remains the same), might lead to a higher risk of adhesive failure. This might be relevant because these fixed retainers are already commercially available and being used on patients.

The results of the present study indicate that the retainer design should always consider the stiffness of the material. For example, multistranded fixed retainers are mostly made from materials like stainless steel or titanium, and their high stiffness has been compensated for by the multistranded geometry. However, the present study demonstrates, that exceeding a certain RD of 0.55 mm leads to a considerable increase in RS and therefore to higher overall stress on the bonding system. Specifying an exact threshold, which is clinically relevant, is not warranted by the data of this study. Therefore, clinical studies have to follow to transfer these results into the clinical context. But considering the correlation between an increase in retainer stiffness and in RD might be relevant when choosing the respective retainer.

The production of fixed CAD/CAM retainers involves laser melting, laser cutting, or milling, which makes creating a multistranded geometry hard to realise. Therefore, the material properties have to be optimised or the diameter has to be adapted according to the material stiffness to achieve the required resilience. However, when choosing materials with lower Young’s moduli, one should remember that these should also provide long-term sufficiency. In this context, flexible NiTi CAD/CAM retainers seemed to be a valuable solution but showed breakage in our previous study [4].

To be able to automate FE model construction with varying retainer dimensions, cylindrical wires were used as retainers. When simulating multistranded fixed retainers, the Young´s modulus of this cylindrical wire was adapted in order to reflect the bending stiffness of the multistranded fixed retainer. As a limitation of this study, this led to an error in the simulated elongation stiffness of multistranded fixed retainers. However, because elongation stiffnesses are much higher than bending stiffnesses of fixed retainers, the effects analysed in this investigation will predominantly be affected by the bending stiffness and this simplification should not lead to erroneous results. Furthermore, the circular bonding area was kept constant for all the fixed retainer designs so that the stress states in the adhesive interfaces could be meaningfully compared. Another limitation was that tooth movements could not be considered with the FE model. The model was also unable to consider retainer failures like fractures and the debonding caused by the failure of the adhesive or a break-out of the retainer from the adhesive.

In summary, the present study investigated the influence of RS and TR on force transmission and stress distribution to inform practitioners which fixed retainer is most suitable in different clinical situations. Higher retainer stiffness, whether caused by a bigger diameter or material properties, increases force transmission to the neighbouring teeth and overall stress in the bonding area, thereby increasing the chance of debonding. A smaller RD leads to a less-uniform stress distribution, which causes local stress peaks that are concentrated to a comparatively small bonding area. Further studies should investigate whether these local stress peaks can cause complete or partial debonding. One should keep in mind that debonding represents only one of multiple possible retainer failures. Therefore, further studies should investigate how the retainer diameter, material properties, and configuration effect the risk of breaking or the ability to retain alignment.

## 5. Conclusions

The present study has shown how fixed retainer stiffness affects force transmission using a new FE retainer model. The results highlight two key messages:Higher bending stiffness, higher tooth resilience, and an oblique load situation all increase force transmission and therefore overall stresses on the bonding area of the loaded tooth and its neighbour teeth.A lower retainer diameter decreases the uniformity of the stress distribution leading to local stress peaks within a small field of the bonding area.In conclusion, using fixed conventional or CAD/CAM retainers with rather low stiffness (up to 3 N/mm) might be recommended, especially in cases with high tooth resilience. Therefore, typical stainless steel multistranded retainers should not exceed a diameter of up to 0.55 mm. However, it is important to note that clinical studies have to validate the results of this FE study as accurate recommendations cannot be made based on a numerical simulation alone.

## Figures and Tables

**Figure 1 bioengineering-11-00394-f001:**
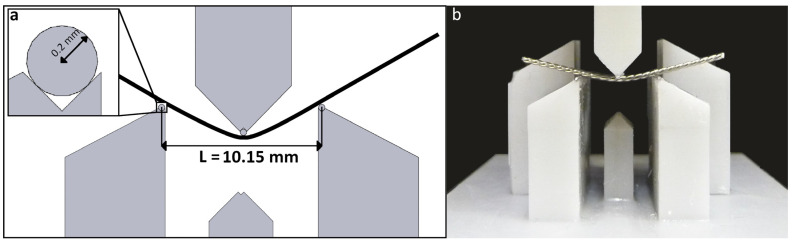
(**a**) Schematic illustration of the three-point bending test setup and (**b**) close-up of the relevant parts with a triple-stranded stainless steel retainer during testing.

**Figure 2 bioengineering-11-00394-f002:**
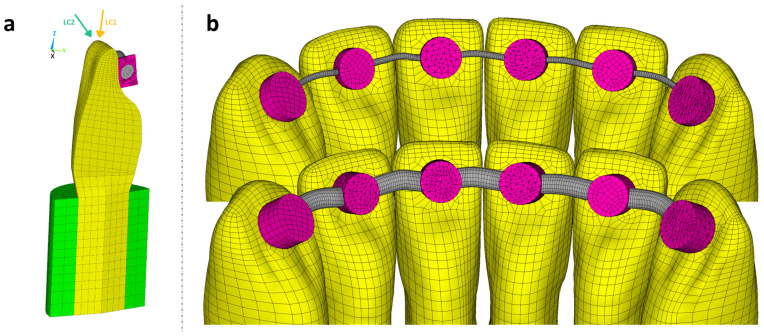
(**a**) Cross section of tooth 31 with all its components including the two load cases and (**b**) close-up of the model with the least and most stiff retainer.

**Figure 3 bioengineering-11-00394-f003:**
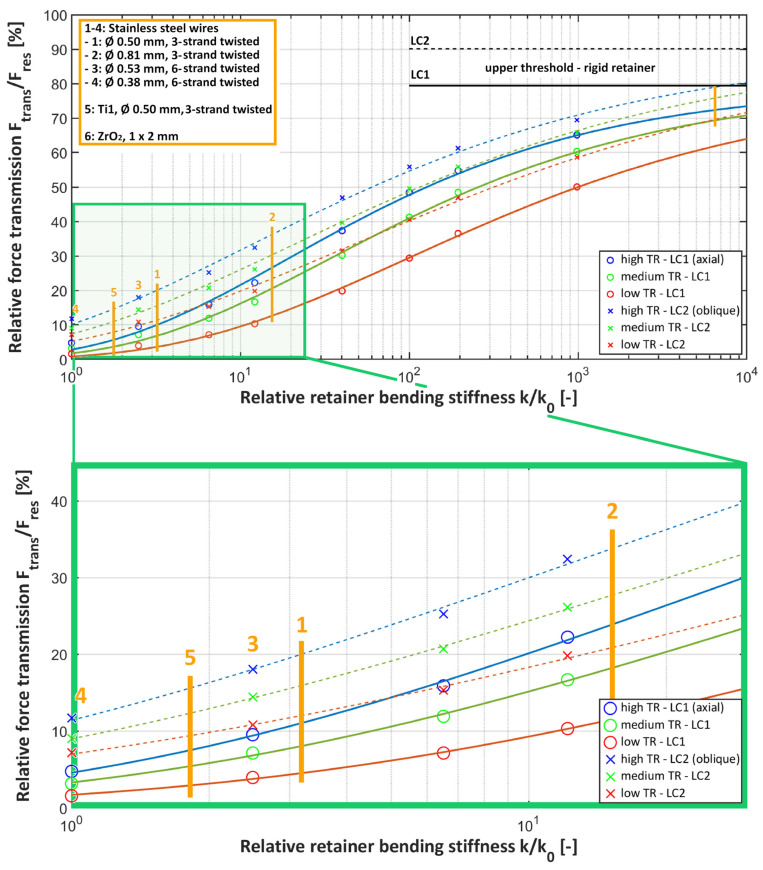
Force transmission (*F_trans_*) to the adhesive bond of the loaded tooth divided by the applied force (*F_res_*) over bending stiffness (*k*), normalised to the least stiff retainer (*k*_0_, no. 4). Data points are fitted with a function of the type seen in Formula (3). The area of conventional hand-bent retainers is highlighted and enlarged with a green rectangle.

**Figure 4 bioengineering-11-00394-f004:**
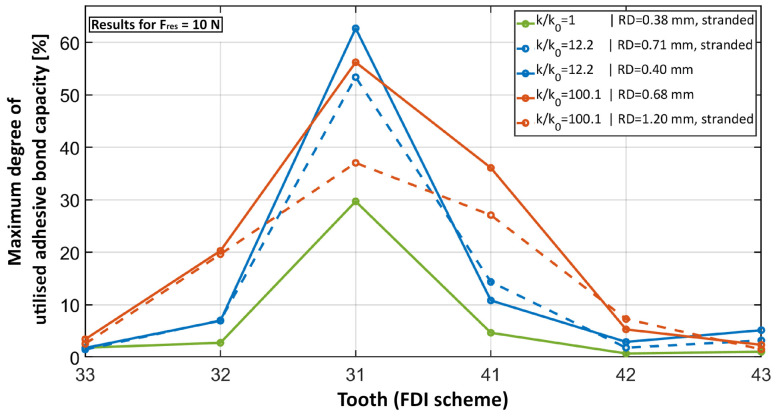
Maximum degree of utilised adhesive bond capacity plotted for each tooth for five different retainers. The retainers demonstrate the effects of a variation in the *RD* configuration. The least stiff retainer (*k*/*k*_0_ = 1) corresponds to the solid green curve. An increase in the *RD* is displayed with the dashed blue curve while the switch to a solid cross section is displayed with the solid blue curve. A further increase in the *RD* is displayed with the dashed red curve while the configuration that is different to the dashed blue curve is displayed with the solid red curve. The solid blue and red curves also demonstrate the effect of an increase in the *RD*. All the displayed models correspond to high TR and LC1.

**Figure 5 bioengineering-11-00394-f005:**
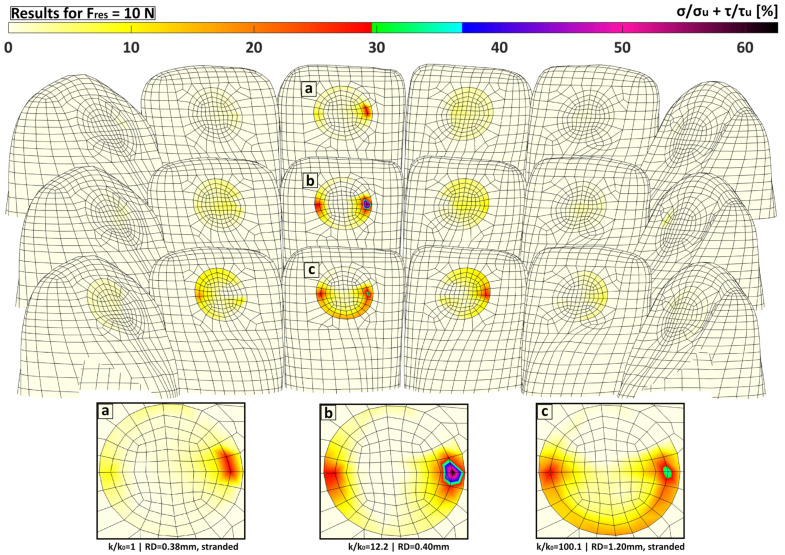
Degree of utilised capacity for three different retainers. The values were calculated according to the fracture hypothesis as described in Formula (4) (*σ*/*σ_u_* + *τ*/*τ_u_*), where σ is the normal stress, *σ_u_* the tensile bond strength, *τ* the resulting shear stress, and *τ_u_* the shear bond strength. (**a**) *k*/*k*_0_ = 1 with *RD* = 0.38 mm in the multistranded configuration, (**b**) *k*/*k*_0_ = 12.2 with *RD* = 0.4 mm and a solid cross section, and (**c**) *k*/*k*_0_ = 100.1 with *RD* = 1.20 mm in the multistranded configuration. (**a**) corresponds to the solid green curve in Figure 4, (**b**) to the solid blue curve, and (**c**) to the dashed red curve. All displayed models correspond to high TR and LC1.

**Figure 6 bioengineering-11-00394-f006:**
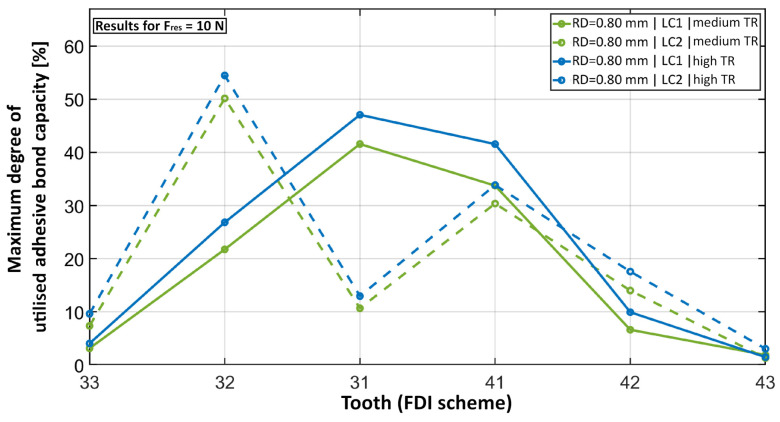
Maximum degree of utilised adhesive bond capacity plotted for each tooth for the same retainer but a different load case (dashed line) and a higher TR (blue).

**Table 1 bioengineering-11-00394-t001:** Details of the retainers included in this study. Retainers with a wide range of diameters were included to cover the range of commercially available retainers.

Number	Retainer Name	Manufacturer	Material	Diameter RD	Configuration
1	Dentaflex	Dentaurum, Ispringen, Germany	Stainless steel	0.50 mm	6-strand twisted, conventional (bendable)
2	Stainless steel lingual retainer	3M Oralcare, Seefeld, Germany	Stainless steel	0.81 mm	3-strand twisted, conventional (bendable)
3	Penta Twist	Gold’n Braces, Tampa, USA	Stainless steel (gold coated)	0.53 mm	6-strand twisted, conventional (bendable)
4	Dentaflex	Dentaurum, Ispringen, Germany	Stainless steel	0.38 mm	6-strand twisted, conventional (bendable)
5	Titanium retainer wire	Dentaurum, Ispringen, Germany	Titanium grade 5	0.50 mm	3-strand twisted, conventional (bendable)

**Table 2 bioengineering-11-00394-t002:** Simulated vertical and horizontal TR correlating with the three defined levels.

Level of TR	Vertical TR [µm/N]	Horizontal TR [µm/N]	Ratio [-]
Low	1.05	4.22	4.02
Medium	2.01	6.04	3.01
High	2.99	9.33	3.12

**Table 3 bioengineering-11-00394-t003:** Overview of the material parameters relevant for the FE analyses.

Material	Young’s Modulus *E* [GPa]	Poisson’s Ratio *ν* [-]
Teeth (dentine)	15	0.30
Adhesive (composite resin)	6	0.30
Retainer	13–200	0.30
Periodontal ligament	Chosen according to the desired TR	

**Table 4 bioengineering-11-00394-t004:** Combinations of simulated *E* and *RD* resulting in the relative retainer bending stiffness (ratio of retainer bending stiffness *k* to the bending stiffness of the least stiff retainer *k*_0_).

Young’s Modulus *E* [GPa]	Diameter *RD* [mm]	Relative Bending Stiffness *k*/*k*_0_
20 *	0.38	1
13 *	0.53	2.5
200	0.34	6.5
200	0.40	12.2
20 *	0.71	12.2
200	0.54	40.1
200	0.68	100.1
20 *	1.20	100.1
200	0.80	194.5
200	1.20	984.7

* Stiffness value and diameter based on a multistranded retainer, use of an equivalent Young’s modulus.

**Table 5 bioengineering-11-00394-t005:** Overview of the three-point bending test results of tested multistranded retainers.

Number	Stiffness *k* [N/mm] Mean Value (SD)	*k*/*k*_0_ [-]
1	2.97 (0.05)	3.1
2	15.80 (0.43)	16.6
3	2.36 (0.04)	2.5
4	0.95 (0.02)	1.0
5	1.76 (0.09)	1.9

**Table 6 bioengineering-11-00394-t006:** Resulting parameters a and b of the fitting function (Formula (3)) associated to each level of TR and each LC. The stiffness of the most resilient retainer was taken as a reference stiffness *k*_0_.

	Low TR	Medium TR	High TR
LC1	a = 4.55|b = 0.33	a = 3.83|b = 0.38	a = 3.34|b = 0.41
LC2	a = 2.85|b = 0.27	a = 2.51|b = 0.31	a = 2.19|b = 0.32

## Data Availability

Further data can be made available to interested parties on request to the corresponding author.

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
