# Peer review of "Finite Element Analysis of Fixed Orthodontic Retainers"

_bioengineering, 2024, doi:10.3390/bioengineering11040394_

Round 1
Reviewer 1 Report
Comments and Suggestions for Authors
The paper “Finite element analysis of orthodontic retainers” used FE model of a lower jaw anterior segment to investigate the effects of Retainer Stiffness on the transmission of force and distribution of stress along adhesive interfaces. The FE model was developed to understand why different retainers have different failure rates.
The article is very interesting, the pictures and the graphs very complete. The FEA model is very well performed.
Nevertheless, there is a major concern:
The paper focused on the bonding between the teeth and the adhesive (To calculate whether the adhesive bond is susceptible to debonding and to illustrate the stress distribution at the bonding interfaces, the normal (σ) and resulting shear stresses (τ) in the bonding interfaces corresponding to a bite force of 10 N were calculated and implemented using a fracture hypothesis (Formula 4) and typical values for the tensile bond strength (𝜎𝑢 = 20 MPa) and shear bond strength (τu = 20 MPa) for the interface between the teeth and the adhesive). This limitation has been outlined among study’s limitation: “One of the major flaws of this paper is the lack of analysis of the bonding failure between the wires and the composite. “
Nevertheless the reviewer thinks that it should be outlined in a more extensive way in the introduction.
Worst failures are not the ones in which the adhesive detaches from the tooth. They are when the bonding between the wire and the adhesive fails and the tooth rotates. This is responsible of many issues among which root exposures and gingival recession are the most important ones.
If this limitation could be clearly outlined and maybe focus on possible future studies, the paper can be accepted since the other investigated analysis is well conducted.
Introduction:
Report that FE is a methodology used in several branches of dentistry (cite 2-3 disciplines such as: (https://doi.org/10.1111/cid.12129, https://doi.org/10.1080/01694243.2017.1304172, https://doi.org/10.1590/0103-6440201300639 )
Author Response
Cover letter to Reviewer 1
- The article is very interesting, the pictures and the graphs very complete. The FEA model is very well performed.
- Response: We thank the reviewer for his kind words which appreciate our work.
- Concern 1: “The paper focused on the bonding between the teeth and the adhesive (…). This limitation has been outlined among study’s limitation: “… “. Nevertheless, the reviewer thinks that it should be outlined in a more extensive way in the introduction. Worst failures are not the ones in which the adhesive detaches from the tooth. They are when the bonding between the wire and the adhesive fails and the tooth rotates. This is responsible of many issues among which root exposures and gingival recession are the most important ones.”
- Response: We highly appreciate the reviewers concern regarding failures between the wire and the adhesive. Our paper focused on the general influence of retainer stiffness and tooth resilience on the force transmission and stress distribution and secondary focused on the bonding failure between the teeth and the adhesive. Therefore, the reviewer is right when saying that this study primarily concentrated on the enamel-adhesive interface. This is primarily because this failure type is most common [1]. Moreover, the state of stress around the wire at the areas where the retainer runs out of the adhesive are not completely equal to the state at the enamel-adhesive interface. However, as described in the discussion of limitations, this would be an interesting topic for further investigations.
We would like to briefly discuss our experience with retainer failures in clinical practice. Apart from the fact, that detachment on the enamel-adhesive interface is the most common failure, we moreover consider this failure type to be particularly critical. This is because they are often not recognized by the patient since the composite spot seem to be intact from the lingual side, only showing a tiny “gap” between the tooth and the adhesive. In these cases, patients only recognize the detachment after subsequent tooth movement which is very troublesome since regular check-ups are often discontinued when failure occurs.
- Revised text: “We concentrated on the enamel-adhesive interface, since detachment in this interface is the most common failure”
- Concern 2: “Introduction: Report that FE is a methodology used in several branches of dentistry (cite 2-3 disciplines such as: (https://doi.org/10.1111/cid.12129, https://doi.org/10.1080/01694243.2017.1304172, https://doi.org/10.1590/0103-6440201300639 )”
- Response: We thank the reviewer for these valuable references. We included all references and added some more on several fields in dentistry.
- Revised text: “Finite element (FE) analysis is an important part of research in many branches of dentistry. Examples include the field of implantology, craniomaxillofacial surgery, endodontics, prosthetics, material research and also of orthodontics. Several studies have investigated retainer failure using finite element (FE) analysis [5-7] and have analysed their bonding behaviour [8,9].”
- Aye, S.T.; Liu, S.; Byrne, E.; El-Angbawi, A. The prevalence of the failure of fixed orthodontic bonded retainers: a systematic review and meta-analysis. Eur J Orthod 2023, 45, 645-661, doi:10.1093/ejo/cjad047.
Reviewer 2 Report
Comments and Suggestions for Authors
The paper has readability issues, revise
Title, what type of retainer are you referring to?, bonded retainer? revise
Abstract, add the main significant finding to the result section, what are the clinical implications? what do you suggest to use, what more, clinical studies?
The conclusion is not good for the reader, remove and revise 'These findings provide insights into how retainer design and material selection affect its long-term stability and therefore emphasises the interplay between biomechanical factors in retainer efficacy. '
introduction
line 44 revise, what do you mean, what type of retainer/set is this, bonded wire ?"In the case of commercially available conventional retainers, the outer diameter can range from 0.38 mm (0.015 inches) to 0.81 mm (0.032 inches), and new CAD/CAM retainers can reach 3.5 mm (0.138 inches) in height.'
line 47, again are you talking about bonded retainers? expand for non-orthodontist readers
line 50, again expand for the reader, most bonded retainer are made of ss or niti or some sort of titanium alloy, ''whereas CAD/CAM retainers can also be made from polyetheretherketone, zirconia (ZrO2), nickel titanium, or cobalt-chromium. These differences directly influence the RS and might therefore affect the transmission of force from a loaded tooth to the neighbouring teeth. '
line 162, 'massive wires with the same diameter' did you mean sold wire
you need to expand more for the general reader terminology such as Maximum degree of utilised capacity
Tables combine the mean and SD and show as Mean(SD)
Conclusion, what type/diameter wire do you suggest based on your findings
Comments on the Quality of English Languageneeds major revision
Author Response
Cover letter to Reviewer 2
- Concern 1: “The paper has readability issues, revise”
- Response: We acknowledge the reviewer's concern. The authors indicate that the manuscript
has been previously revised by a professional English language editor and the supporting document has been uploaded to the system.
- Concern 2: “Title, what type of retainer are you referring to?, bonded retainer? Revise”
- Response: We thank the reviewer for this important question, which we want to address by explaining the classification of retainers in more detail from an orthodontic point of view. Regarding retainers, there are two main groups: removable retainers (usually plates or removable splints) and fixed (bonded) retainers. However, fixed (bonded) retainers are shown to be more effective in preserving the previously achieved alignment in the anterior region (https://pubmed.ncbi.nlm.nih.gov/30075919/) and are therefore commonly used in everyday clinical practice for decades. Indeed, our investigation centred on fixed (bonded) orthodontic retainers. Since the term "fixed" is the common term in orthodontic terminology, we have changed the title accordingly. However, if the reviewer considers the term "bonded" to be more appropriate, we can of course change the title accordingly. We moreover added the term “fixed” in multiple passages in the revised manuscript, to guarantee a better understanding for a broader readership.
- Revised text: Finite element analysis of fixed orthodontic retainers.
- Concern 3: “Abstract, add the main significant finding to the result section, what are the clinical implications? what do you suggest to use, what more, clinical studies?”
- Response: We thank the reviewer for bringing these points to our intention! Due to the results of our study we strongly recommend to avoid fixed retainers with excessively high stiffness. We would like to briefly describe the development of fixed retainers in recent years – Over decades, orthodontic fixed retainers were designed with the idea of reducing the amount of stiffness by creating designs like the so called “multistranded” geometry. Due to new technology capabilities, CAD/CAM retainers from stiff materials and in bulky designs were introduced during the last ten years, in some cases infringing the principle of a rather low stiffness. Due to the results of our study, we are convinced that the need for a rather low degree of stiffness should be noticed again. Not only because of dental health, but also because of the long-term stability of the retainers. Since this is an FEM study, precise suggestions regarding the “best” or “most suitable” retainer are not strengthened by the results of our study. However, we fully agree that the recommendation for retainers with rather low amounts of stiffness had to be added. We moreover added the key findings of our investigations (described in Concern 4).
- Revised text: “Additionally, a smaller retainer diameter reduced the uniformity of the stress distribution in bonding interfaces causing concentrated stress peaks within a small field of the bonding area. An increase in retainer stiffness and in tooth resilience as well as a more oblique load direction all lead to higher overall stress in the adhesive bonding area associated with a higher risk of retainers bonding failure.”
- Concern 4: “The conclusion is not good for the reader, remove and revise 'These findings provide insights into how retainer design and material selection affect its long-term stability and therefore emphasises the interplay between biomechanical factors in retainer efficacy.”
- Response: We fully agree that the previous conclusions in the abstract section had to be revised since they were not informative, not concluding the key findings of our investigation. Therefore, we changed them to
- Revised text: “Additionally, a smaller retainer diameter reduced the uniformity of the stress distribution in bonding interfaces causing concentrated stress peaks within a small field of the bonding area. An increase in retainer stiffness and in tooth resilience as well as a more oblique load direction all lead to higher overall stress in the adhesive bonding area associated with a higher risk of retainers bonding failure. Therefore, it might be recommended to avoid the use of retainers that are excessively stiff, especially in cases with high tooth resilience.”
- Concern 5: “introduction… line 44 revise, what do you mean, what type of retainer/set is this, bonded wire ?"In the case of commercially available conventional retainers, the outer diameter can range from 0.38 mm (0.015 inches) to 0.81 mm (0.032 inches), and new CAD/CAM retainers can reach 3.5 mm (0.138 inches) in height.”
- Response: The investigation was for fixed orthodontic retainers. Like described in concern 2, we changed the whole text to the term “fixed” retainer, where appropriate in order to guarantee better understanding for a broader readership.
- Revised text: Not applicable.
- Concern 6: “line 47, again are you talking about bonded retainers? expand for non-orthodontist readers”
- Response: We thank the reviewer for bringing this point to our intention. Like described in concern 2, we checked and revised the whole manuscript accordingly.
- Revised text: Not applicable.
- Concern 7: “line 50, again expand for the reader, most bonded retainer are made of ss or niti or some sort of titanium alloy, ''whereas CAD/CAM retainers can also be made from polyetheretherketone, zirconia (ZrO2), nickel titanium, or cobalt-chromium. These differences directly influence the RS and might therefore affect the transmission of force from a loaded tooth to the neighbouring teeth.”
- Response: We thank the editor for his input and expanded the corresponding -paragraph.
- Revised text: “Regarding the retainer material, conventional retainers are made from stainless steel, gold, grade 1 titanium, grade 5 titanium, or titanium-molybdenum, whereas CAD/CAM retainers can also be made from polyetheretherketone, zirconia (ZrO2), nickel titanium, or cobalt-chromium. The material selection directly influences the RS due to the resulting differences in Young's Modulus and might therefore affect the transmission of force from a loaded tooth to the neighbouring teeth.”
- Concern 8: “line 162, 'massive wires with the same diameter' did you mean sold wire”
- Response: We thank the reviewer for their input and will use solid instead of massive throughout the manuscript.
- Revised text: Not applicable.
- Concern 9: “you need to expand more for the general reader terminology such as Maximum degree of utilised capacity”
- Response: We appreciate the reviewers input and therefore expanded section 2.3, where the terminology degree of utilised capacity is defined. Moreover, we expended section 3.3 to elaborate on the terminology of maximum degree of utilised capacity.
- Revised text:
- 3: “To calculate whether the adhesive bond is susceptible to debonding and to illustrate the stress distribution at the bonding interfaces, the normal (σ) and resulting shear stresses (τ) in the bonding interfaces corresponding to a bite force of 10 N were calculated and implemented using a fracture hypothesis (Formula 4) and typical values for the tensile bond strength (u = 20 MPa) and shear bond strength (τu = 20 MPa) for the interface between the teeth and the adhesive [13-16]. The resulting value represents the utilised capacity of the adhesive bond for the corresponding combination of RS, TR, and load case. A high degree of utilised capacity in a certain area therefore corresponds to high normal and resulting shear stresses in that area”.
- 3: “While the overall amount of force transmitted to the bonding area of the loaded tooth increased with increasing RS, the utilised capacity calculated with Formula (4) did not necessarily increase as well because of the non-uniformity of the stress distribution (Figure 4 and Figure 5). Figure 4 presents the maximum degree of utilised capacity for selected retainers that illustrate the effects of an increasing RS due to the increase in RD and/or the use of a solid cross section instead of a multistranded configuration. The resulting value corresponds to the highest amount of ultilised capacity that occurs at each tooth respectively. Within the region of conventional hand bent retainers (comparing k/k0 = 1 and k/k0 = 12.2), the increase in RS (here by a factor of 12.2) increased the utilised capacity. Although this was the case for both the increase in RD (dashed blue curve in Figure 4) and a switch to the solid cross section (solid blue curve), the maximum degree of utilised capacity increased less with the increase in RD.”
- Concern 10: “Tables combine the mean and SD and show as Mean(SD)”
- Response: We acknowledge the reviewers input and therefore adapted table 5
- Revised text: Not applicable.
- Concern 11: “Conclusion, what type/diameter wire do you suggest based on your findings”
- Response: Like we discussed in Concern 3, suggestions regarding the exact type/diameter are not strengthened by the results of this investigation. However, we agree that suggestions regarding a rather low stiffness and a medium diameter had to be included within the Conclusions to draw the clinician's attention to this interrelation.
- Revised text: “In conclusion, using retainers with rather low stiffness and medium diameter might be recommended, especially in cases with high tooth resilience. “

Round 2
Reviewer 1 Report
Comments and Suggestions for Authors
The authors provided all the requested improvements
Author Response
We would like to thank the reviewer for his/her contribution and acknowledgement of the revised manuscript.
Reviewer 2 Report
Comments and Suggestions for Authors
thank you for the revisions,
conclusion, please add some values in this sentence'n conclusion, using retainers with rather low stiffness and medium diameter might be 349
recommended, especially in cases with high tooth resilience'
needs revision
